# Evaluating the Piezoelectric Energy Harvesting Potential of 3D-Printed Graphene Prepared Using Direct Ink Writing and Fused Deposition Modelling

**DOI:** 10.3390/polym16172397

**Published:** 2024-08-23

**Authors:** Hushein R., Thulasidhas Dhilipkumar, Karthik V. Shankar, Karuppusamy P, Sachin Salunkhe, Raja Venkatesan, Gamal A. Shazly, Alexandre A. Vetcher, Seong-Cheol Kim

**Affiliations:** 1Vel Tech Rangarajan Dr. Sagunthala R&D Institute of Science and Technology, Chennai 600062, India; husheinece@gmail.com; 2Department of Mechanical Engineering, Amrita Vishwa Vidyapeetham, Amritapuri 690525, India; karthikvs@am.amrita.edu; 3Centre for Flexible Electronics and Advanced Materials, Amrita Vishwa Vidyapeetham, Amritapuri 690525, India; 4Department of Chemistry, Vinayaka Mission’s Kirupananda Variyar Engineering College, Vinayaka Mission’s Research Foundation (DU), Salem 636308, India; karuppusamy98422@gmail.com; 5Department of Mechanical Engineering, Gazi University, 06560 Ankara, Turkey; sachinsalunkhe@gazi.edu.tr; 6School of Chemical Engineering, Yeungnam University, 280 Daehak-ro, Gyeongsan 38541, Republic of Korea; sckim07@ynu.ac.kr; 7Department of Biomaterials, Saveetha Dental College and Hospitals, SIMATS, Saveetha University, Chennai 600077, India; 8Department of Pharmaceutics, College of Pharmacy, King Saud University, P.O. Box 2457, Riyadh 11451, Saudi Arabia; gaahmed@ksu.edu.sa; 9Institute of Biochemical Technology and Nanotechnology, Peoples’ Friendship University of Russia n.a Lumumba (RUDN), 6 Miklukho-Maklaya St., 117198 Moscow, Russia; avetcher@gmail.com

**Keywords:** energy harvesting, 3D printing, direct ink writing (DIW), fused deposition modelling (FDM), graphene

## Abstract

This research aims to use energy harvested from conductive materials to power microelectronic components. The proposed method involves using vibration-based energy harvesting to increase the natural vibration frequency, reduce the need for battery replacement, and minimise chemical waste. Piezoelectric transduction, known for its high-power density and ease of application, has garnered significant attention. Additionally, graphene, a non-piezoelectric material, exhibits good piezoelectric properties. The research explores a novel method of printing graphene material using 3D printing, specifically Direct Ink Writing (DIW) and fused deposition modelling (FDM). Both simulation and experimental techniques were used to analyse energy harvesting. The experimental technique involved using the cantilever beam-based vibration energy harvesting method. The results showed that the DIW-derived 3D-printed prototype achieved a peak power output of 12.2 µW, surpassing the 6.4 µW output of the FDM-derived 3D-printed prototype. Furthermore, the simulation using COMSOL Multiphysics yielded a harvested output of 0.69 µV.

## 1. Introduction

The current era heavily focuses on technological development and innovation in the energy sector. There has been a notable shift towards prioritising green and renewable energy systems in the 21st century [1]. The emergence of new materials with strong ecological properties has influenced this shift. Additionally, the progression from coal to oil as a primary fuel source to the dominance of silicon-based computers has also been observed [2,3,4]. The discovery of nano-materials is expected to advance renewable energy technology further. Furthermore, the urgent need for new materials is not just for upgrading energy systems but also for energy harvesting solutions [5].

Additive manufacturing, also known as 3D printing, creates three-dimensional objects directly from a computer-aided design (CAD) model by adding layers [6]. Unlike traditional manufacturing, which involves extensive planning and specific tooling, additive manufacturing requires only basic dimensional information about the part, machine, and materials [7]. It allows for the creation of intricate shapes and has led to faster, smaller, and more affordable printing methods. Overall, 3D printing methods adhere to the fundamental principles of additive manufacturing [8]. According to ASTM 52900 [9], additive manufacturing [10,11,12] can be classified into seven categories:(i)Vat photopolymerization;(ii)Material extrusion;(iii)Material jetting;(iv)Binder jetting;(v)Powder bed fusion;(vi)Direct energy deposition;(vii)Sheet lamination.

Fused deposition modelling (FDM) is an additive manufacturing method that involves gradually extruding melted plastic filament through a nozzle to create models layer by layer. This technique is versatile and can be used to produce prototypes, functional items, and porous structures [13,14,15]. On the other hand, Direct Ink Writing (DIW) relies on ink extrusion without the need for a material bed for support. DIW is particularly effective in fabricating porous structures, such as tissue engineering scaffolds [16]. Graphene’s unique structure, consisting of a two-dimensional carbon sheet with sp^2^ hybridisation, allows it to maintain a thickness of just one atom. The carbon atoms in graphene are arranged in a honeycomb-like network, forming a lattice structure [17,18,19,20]. Graphene is often referred to as the “mother” of all graphite materials. One of the key features of graphene is its extensive π-conjugation, which gives it the characteristics of a semimetal or a semiconductor with a zero-gap level. This property enables graphene to exhibit exceptional electrical properties, including the peculiar quantum Hall effect and high electron mobility at room temperature. In fact, the electron mobility of graphene is approximately 200,000 cm^2^V^−1^S^−1^, surpassing that of carbon nanotubes [21].

Fused deposition modelling (FDM) and Direct Ink Writing (DIW) are prominent 3D printing techniques, each with its strengths and limitations. FDM, while accessible and versatile, often suffers from layer lines, inconsistent mechanical properties, and limited material options. Its reliance on thermoplastic filaments restricts its application in certain industries. Conversely, DIW offers greater material flexibility, enabling the printing of complex geometries and functional components. However, it can be slower and more expensive and requires precise material formulation [22].

Both FDM and DIW face challenges in achieving high-resolution, isotropic properties and large-scale production efficiency compared to other 3D printing methods like Stereolithography (SLA) and Selective Laser Sintering (SLS). SLA excels in producing highly accurate parts with smooth surfaces, but material choices and post-processing requirements limit it. SLS offers broader material compatibility and build volume but results in lower surface quality. Ultimately, the optimal 3D printing method depends on the specific application, considering factors like material properties, part complexity, production volume, and cost-effectiveness. So, based on this, it is concluded that FDM and DIW have been considered the best ones to provide better results [23].

Graphene is composed of a single layer of carbon atoms in a 2D structure and exhibits impressive multifunctional characteristics [24]. Its outstanding strength, thermal and electrical conductivity, and chemical reactivity contribute to its high versatility. Although graphene lacks inherent piezoelectricity, it can demonstrate piezoelectric characteristics as a result of adsorbed atoms on its surface. This unique property finds utility in chemical detection, energy generation, and high-frequency sound manipulation. Morozovska et al. [25] examined the piezoelectric effect in ferroelectric substrates impacts graphene conductance, while Rodrigues et al. [26] demonstrated how single-layer graphene on SiO_2_ grating substrates has a high piezo coefficient. Overall, these findings highlight the potential of graphene’s piezo-conductive properties for a range of future applications. Recent studies have delved into the potential of graphene-based materials in energy harvesting and storage applications through the use of 3D printing methods. Fused deposition modelling (FDM) has been employed to produce graphene-based polylactic acid filaments for 3D printing electrodes, which have exhibited potential as anodes in Li-ion batteries and as catalysts for hydrogen evolution reactions [27,28].

High-quality graphene with over 99% purity was used for this study. Graphene of high quality, which is defined by a low defect density and substantial flake size, generally demonstrates enhanced electrical conductivity, a vital attribute for effective energy conversion. In the context of piezoelectric energy harvesting, graphene exhibiting a high piezoelectric coefficient is preferred. Ultimately, the selection of the most appropriate graphene grade for a given energy harvesting application necessitates a meticulous consideration of these factors alongside the required performance criteria. High-quality graphene is generally favoured for energy harvesting purposes; the specific demands of each application determine the most appropriate grade. Considerations such as the type of energy harvesting (e.g., piezoelectric or thermoelectric), the desired power output, and the feasibility of production scalability are essential when choosing graphene to achieve optimal performance.

Mallineni et al. [29] demonstrated a wireless triboelectric nanogenerator using graphene–PLA nanocomposite, achieving high output voltages and wireless energy transmission. Ben Achour et al. [30] investigated uniaxially stretched PLA films for piezoelectric energy harvesting, showing potential comparable to PVDF. Oumghar et al. [31] reported enhanced piezoelectric properties in graphene oxide–PLA nanocomposite films, attributing this to the presence of the β-polymorph. These studies highlight the potential of graphene–PLA composites for energy harvesting applications, offering advantages such as eco-friendliness, low cost, and flexibility. Amini et al. [32] created a low-cost, flexible PVA-based triboelectric nanogenerator capable of powering small electronics and sensors. Wu et al. [33] enhanced the piezoelectric properties of polyvinylidene fluoride (PVDF) films by incorporating reduced graphene oxide, achieving significant improvements in open-circuit voltage and harvested power. These studies highlight the potential of graphene and PVA-based materials in energy harvesting devices, offering promising solutions for self-powered electronics, sensors, and IoT applications.

The incorporation of graphene into ABS filaments has been shown to enhance the mechanical properties of FDM-printed components. Furthermore, graphene-based nanogenerators have been explored for energy harvesting applications, offering the possibility of self-powered portable devices, as shown by Kwon et al. [34]. These investigations illustrate the adaptability of graphene in 3D-printed energy systems, underscoring its capacity to enhance conductivity, mechanical resilience, and energy harvesting capabilities across a range of applications. The central theme of this proposed method is to print graphene material using an extrusion-based method and analyse the performance of a 3D-printed model using both simulation and experimental procedures [35,36]. The step-by-step procedures for the overall process are as follows. Graphene has been used as the primary printing material in this study. The proposed 3D model was developed using a mechanical design platform, AUTOCAD 2018. The final designed model is converted into STL format for the smooth completion of the 3D printing process. The dog bone shape model has been finalised for the proposed 3D model [37,38,39,40]. The primary purpose of choosing the model is to obtain good mechanical vibrations. This study seeks to assess the energy harvesting capabilities of a 3D-printed model created using Direct Ink Writing (DIW) and fused deposition modelling (FDM) techniques [41,42,43]. The research involves the examination of 3D-printed graphene-based composites through both simulation and experimental methods. The simulation was conducted using the COMSOL Multiphysics 5.6a tool, while the experimental analysis utilised a vibration-based Cantilever beam technique [44,45].

## 2. Materials and Methods

### 2.1. Proposed Design

The dimensions of the proposed model are shown in Figure 1. A small structure size of about 70 mm in length, 15 mm in width, and 3 mm in thickness was considered for this proposed work.

### 2.2. Materials

Generally, graphene material is available in powder form and must be prepared in the required form per the printing methodology used. The graphene slurry needs to be prepared for the DIW printing process, and graphene wire needs to be prepared for the FDM printing process [46,47]. For both the DIW and FDM process, the graphene material was purchased from Blackmagic 3D by Graphene 3D Lab, located in New York, the United States. For the FDM process, graphene with a PLA spool is readily available. So, the spool is directly used in this FDM process for printing. In the extrusion-based 3D printing technique, the extrusion nozzle only determines the shape and size of the filament [48,49]. The air pressure unit controls the material flow through the nozzle. The general extrusion-based 3D printing process is shown in Figure 2.

### 2.3. Preparation of 3D-Printed Graphene-Based Composite Using the DIW Process

The material preparation process for the DIW process involved two steps. The first step required preparing a polyvinyl alcohol (PVA) solution, while the second step involved creating a graphene slurry [50,51]. The slurry was made by mixing 20 wt.% of the PVA solution with 80 wt.% of graphene powder. Following this, a 3D design model was produced using AUTOCAD and then converted into a digital approximation (.STL file). The digital model was then sliced into layers, and a programming code for the printing process (G-Code) was generated. Subsequently, the file was transferred to the 3D printer [52,53,54]. Finally, the proposed 3D design model was printed layer by layer. The entire process is described in Figure 3.

The slicing process is the most crucial part of the printing process, where the printing parameters have been defined for printing the 3D design of the component to be developed. In this proposed work, Ultimaker Cura 5.4 software was used to convert the 3D design component from STL format to G-code format for the 3D printer [28,55,56]. The overall process involved in the DIW printing process is illustrated in Figure 3A. Once the experimental printing process was completed, the designed model was printed with the expected outcomes. The exact dimension is included in the printed model. Figure 3B shows the printed model.

### 2.4. Preparation of 3D-Printed Graphene-Based Composite Using the FDM Process

In current 3D printing manufacturing technology, fused deposition modelling is considered one of the best and most valuable technologies. The filament is melted and deposited onto a build platform through a nozzle, where it solidifies. This method obtains this remarkable position over other methods. The raw materials are accessed in filaments, and then the extruder ensures that the material is easily pushed through the hot end. Like DIW, this method also has the freedom in material selection with a variety of properties, and it ranges from stiff to flexible [57,58,59,60,61]. The central theme of the FDM process is to create the 3D model by creating the 3D structure layer by layer from bottom to top by supplying the semi-molten filament. Figure 4 illustrates the experimental process of the FDM-based 3D printing process.

This proposed work considers the composition of graphene filament with polylactic acid (PLA) for the printing process. The filament has the following primary properties [11,28,39]:Volume resistivity: 0.6 ohm-cm;Colour: black;Diameter: 1.75 mm;Weight: 100 g;Graphene for superior conductivity and improved mechanical properties;PLA-based.

The volume resistivity of the given filament is 0.6 ohm-cm, and it measures the material’s resistance within the cubic centimetre by constantly providing a filament to a nozzle heated at a desired temperature just above the filament melting point [62,63]. From that point, semi-melted material is extruded through the moving controlled nozzle to create the desired 3D structure [64,65]. After the formation of the layer, the filament follows from the previously deposited layer and immediately cools after allowing its solidification. When the placed layer is solidified with a predefined 3D design pattern, each layer reduces the distance equal to the thickness of the layer until the final model completes the printing process. The graphene material spool brought from Black Magic 3D was fixed into the 3D printer and, based on the printing parameters, was extruded at approximately 180 °C. A 0.4 mm nozzle is used to print the required design model. All of the process parameters were assigned to the flash print with Ultimaker Cura 5.4 software to control the printing flow. During the printing process, the printer’s temperature is set to 50 °C for the build platform and 210 °C for extrusion. Figure 5 shows the FDM-based 3D printer with graphene material and the processed output 3D-printed model.

## 3. Testing Approaches

### 3.1. Microstructural Analysis

The microstructure of the 3D-printed graphene sample is determined using SEM analysis. Using a scanning electron microscope (SEM) is a primary method for characterising material structures at a micro-scale. For this experiment, we have used the FEI-Quanta FEG 200F SEM machine Hillsboro, OR, USA at IIT Madras. In this study, an SEM was employed to analyse the surface fractures, corrosion, and potential structural defects of printed graphene using both 3D printing techniques. The electron source emits a stream of electrons that interact with the sample under analysis.

### 3.2. Experimental Analysis of Vibration-Based Energy Harvesting Method

The experimental procedure was performed with a consideration of simulation input parameters and conditions. The electrical signal is generated from the graphene material based on the vibrations [66]. As per the block diagram in Figure 6, the components are connected and made ready for the experiment. The printed component based on DIW is used for the process. Once the process has been completed, the same process will be followed for the FDM-based 3D-printed component. The printed material is mounted on the shaker, which is considered a vibration source. The frequency generator generates the frequency signal, and based on this shaker, the vibrations are produced. The printed device is mounted on a vibration source. So, it drives the cantilever beam structure to obtain the vibration. The accelerometer monitors the acceleration and amplitude of vibration, which is connected to the vibration source [37,67].

The cantilever beam setup was calibrated properly before being used for the experiment. The procedure encompasses the establishment of measurement parameters, the calibration of excitation and measurement systems, the optional execution of modal analysis, an assessment of frequency response functions, the creation of calibration curves, and an evaluation of repeatability and accuracy. This methodology guarantees a precise relationship between the input excitation and the resultant measured output response [68,69,70,71,72].

A multimeter is connected to the printed graphene device, and the required voltage is measured across the top and bottom layers of the graphene layer through lead wires. The expected output voltages of the printed device at different vibration conditions were systematically measured. The shaker vibrations are based on the frequency signal generated by the frequency generator. The accelerometer monitors the vibration, and it looks like the formation of the sudden rise and, after a few fluctuations, comes to a normal position. This cycle appears continuously one by one, so the printed graphene is subject to a continuous change.

### 3.3. Simulation

The proposed research work is focused on the energy harvesting analysis of 3D-printed graphene. The simulation of energy harvesting performance analysis is carried out using COMSOL Multiphysics, and experimental techniques were performed using the cantilever beam-based vibration energy harvesting method. The COMSOL Multiphysics tool is used to model the 3D design model [66]. For this analysis, some specific parameters were chosen for performing the meshing analysis, such as a number of vertex elements, edge elements, and boundary elements. Based on the (AUTOCAD 2018), 3D CAD design, these were analysed. The same set of data was been verified for the vibration analysis. The meshing analysis performed in COMSOL Multiphysics is described below in Table 1.

Three modules were designed, namely solid mechanics, heat transfer, and AC/DC modules. All these modules mentioned above helped to model the graphene-based 3D design structure and run the model simulation in the virtual platform [67]. In the initial condition, the modelling of the substrate is carried out, which improves the material property. The steps for the modelling and simulation of the graphene-based 3D design structure are shown in Figure 7. COMSOL Multiphysics software tool provides a standard virtual platform for studying the properties of the materials, examining the input parameters and analysing the effects of variation in the geometrical values.

The modelling of the substrate is the initial step that starts the entire process. Once the substrate has been finalised for modelling, the sensing elements are fixed. These elements help the system to determine the required conditions for conductive property and energy harvesting analysis for the simulation [69]. Based on the input parameters and conditions, the simulation was initialised to perform the analysis. Once the simulation was performed for a single iteration, the results obtained were compared with previous iteration results [73].

## 4. Results and Discussion

### 4.1. Experimental Analysis of Cantilever Beam Technique

#### 4.1.1. Voltage and Frequency

When frequency variations occur, the material property reacts to these changes, generating the voltage signal. When it achieves the maximum fluctuation, it generates the maximum voltage. Based on the material properties, reactions for the maximum fluctuation give the maximum voltage. The DIW- and FDM-based printer model shows a similar level of variation in the voltage output. However, compared with FDM, the DIW voltage output is high. This analysis is shown in Figure 8.

The output voltage and power were recorded under different frequencies, and load and acceleration conditions were obtained through the experiments. The variance in the experimental output with the simulation output is because of changes in some factors in dielectric loss, coupling loss, and transmission loss, which were not considered in the simulation. The printed graphene material is typically not identical to ideal graphene, so ideal graphene parameters are used as input for the simulation experiments. The experimental results are then compared to the simulation results to determine any differences. Through this comparison, it was found that there were only minor discrepancies in the aforementioned factors. This result concludes that the output from the experiment and simulation results were almost identical. So, the results were acceptable. This indicates that the 3D-printed graphene has better power generation performance. Likewise, Haque et al. [74] describes the development of a 3D-printed triboelectric device capable of harvesting energy and detecting mechanical deformations, with a maximum power density of 10.6 μW/cm^2^. Bhavanasi et al. [75] states that bilayer films containing poled PVDF-TrFE and graphene oxide exhibit better piezoelectric energy harvesting performance, exhibiting a voltage and power output of 4.41 μW/cm^2^. Based on this comparison, this study showed better energy harvesting capabilities. Karan et al. [76] describes how an Fe-doped reduced graphene oxide/PVDF nanocomposite film exhibits excellent piezoelectric energy harvesting performance. The primary outcomes measured in this study were the open-circuit output voltage (up to 5.1 V) and short-circuit current (up to 0.254 μA) of the Fe-doped RGO/PVDF nanocomposite film when subjected to repetitive human finger imparting. Kwon et al. [77] states that graphene transparent electrodes improve the performance of PZT-based piezoelectric energy harvesters. The primary outcome measured in this study was the performance of the PZT nanogenerator, specifically the output voltage (~2 V), current density (~2.2 μA cm^−2^), and power density of ~88 mW cm^−3^.

#### 4.1.2. Power and Frequency

Figure 9A illustrates a comprehensive study of power and frequency changes with various load resistance conditions. In this analysis, 50 kΩ, 100 kΩ, and 500 kΩ were used for the study. It shows the power and frequency analysis with the various resistive load conditions of the DIW-based 3D-printed model. In all these load conditions, the frequency mode started to increase towards the maximum power, and once it reached the maximum value, it gradually decreased to the minimum value. The printed model has a composition of graphene and PVA. It gives the maximum mechanical stability. In addition to the graphene electrical features, it provides the output power based on the different frequencies at various load conditions, as mentioned earlier.

At low-range load conditions, the printed model also shows variations in the vibrations. Due to those vibrations, the printed model generated power. Initially, the power value did not change under the initial frequencies; once it reached 309 Hz, sudden changes occurred, and the power value moved towards the maximum range when it reached the frequency value of 342 Hz. Then, a sudden decrease occurred and reached the lowest power value at 376 Hz. Finally, for some frequencies, it just maintained a low power value. The maximum power value reached 12.22 µW, 6.11 µW, and 1.24 µW for the load conditions of 50 kΩ, 100 kΩ, and 500 kΩ, respectively. When the load resistance keeps decreasing, the power appears to keep increasing. The same changes occurred in the FDM-based 3D-printed model, which is shown in Figure 9B. The same level of variation appeared in the power and frequency analysis. When the frequency reached 303 Hz, the power value started to increase, and at 338.89 Hz, the power value reached the maximum of 6.4 µW, 3.2 µW and 0.6 µW for the various resistive load conditions of 50 kΩ, 100 kΩ, and 500 kΩ, respectively. When the frequency reaches 372.45 Hz, a low power value is maintained.

An analysis of the experimental output power and load resistance of both DIW- and FDM-based 3D-printed structures is shown in Figure 10. In both DIW- and FDM-based 3D-printed models, the harvested power output varied in terms of the variable resistance load. In DIW, the harvested power output reached the maximum value of 31.86 µW at the resistance range of 311.6 kΩ. However, the harvested output power in FDM reached 22.09 µW at the variable resistance range of 318 kΩ.

Table 2 compares the harvested power output at different frequencies obtained from the different load conditions. Both printed models achieved the highest harvested power output at minimum load conditions (50 kΩ). Both 3D-printed models showed the same level of power and frequency analysis variation. Due to their PVA content, DIW-based 3D-printed structures have good mechanical stability. So, they receive the maximum vibration and achieve a good power value in all of the resistive load conditions. Finally, it is concluded that the DIW-based 3D-printed model has achieved maximum harvested power.

### 4.2. Microstructure Analysis

#### 4.2.1. SEM Analysis

A scanning electron microscope (SEM) is used to analyse the morphology of material structures. In this proposed work, SEM images were used to analyse the surface fractures, corrosion, and possible structural defects of graphene material, as shown in Figure 11. In both FDM and DIW printed models, the SEM data show that graphene has a wide and thick particle size indicated at 200 µm magnification. The distance between the graphene particles looks wider with a far position, as shown in Figure 11A,B. The data show the existence of many stacks in the printed graphene structure and indicates that graphene has a layered structure. The DIW SEM image shows that it has a smaller particle size and formed aggregates that are equally spread in the entire surface area. It indicates that fewer stacks exist and that the structure has been exfoliated on graphene. The FDM SEM image shows that the graphene has a rippled surface.

The uniform size of particles appeared on the surface area, and it appeared in 200 µm magnification. Graphene materials are randomly arranged with a thin layer and are tightly linked to one another to form a regular solid surface. However, the obtained graphene structure appears to be multi-layered based on SEM images. This indicates a smoother build-up area, and the visible reduction in the structure suggests exfoliation.

#### 4.2.2. XRD Analysis

The X-ray diffraction (XRD) technique was used to analyse the crystal structure of graphene. The interlayer spacing and distance between carbon atoms in the graphene planes increased from the peak value. As shown in Figure 12, the XRD pattern showed strong characteristic peaks at 2Ɵ = 17.63° for Direct Ink Writing (DIW) and 2Ɵ = 31.63° for fused deposition modelling (FDM), corresponding to a lattice spacing of d = 3.34410 Å. Graphene exhibited a highly crystalline hexagonal lattice with P63/mmc space group symmetry. Notably, the d-spacing of graphene increased due to the influence of 3D printing techniques, while the successful reduction of graphene oxide led to decreased d-spacing. Interestingly, no crystalline peaks were found in the graphene composition, and DIW showed fewer peaks compared to FDM, resulting in weaker interlayer interactions within the graphene composite. This behaviour is attributed to the irregular structure formed by the dilution effect of the polymer matrix.

### 4.3. COMSOL Analysis

Meshing is the most important step in properly simulating a model. In the meshing process, the entire model is separated into non-overlapping elements [19,72]. Separated elements are analysed independently, and the final simulation results are grouped with collections of all the element’s responses. Finally, the perfect mesh is considered for analysing the minimal elements of the designed structure.

Figure 13A shows the meshed surface of the graphene-based designed structure. The coarse mesh analysis is executed with the aid of the analytical value involved in the process and to perform the Grid Independence test. As per methodology, an excellent mesh structure is selected for the final output analysis. Using the final output, a proper analysis of the geometry of the 3D-designed model is performed. The COMSOL Multiphysics simulation of the graphene-based 3D-printed designed structure is shown in Figure 13B. The simulating image uses graphene as a sensing element, and the stress has been generated in the order of 10 power values. The simulation image of von Mises stress and total displacement are mentioned separately.

The total displacement simulation image shows how the displacement occurred in the entire model concerning the pressure. The variation in the pressure makes a displacement in the entire model. Applying the load on the surface of the model at particular points, the model is displaced from its initial position, and it is noted that the maximum and minimum displacements occur. The variation in the displacement value is shown in Figure 13C. The maximum displacement occurred in the load applied area, which is mentioned with the red colour bar. The colour bar clearly shows the minimum and maximum stress variation in the order of 10 power values, indicating how the structure responds to the vibration force.

In both simulating images, the red colour shows the maximum stress and displacement at the boundaries, and the edges, which are very near the load area, measure the maximum stress. Moreover, the other area measures the minimum stress. Moreover, a minimum stress value is obtained at the mid-position between the centre and the edges around the places. The global definitions of parameters are set under the variance in load resistance and acceleration to analyse displacement and voltage with frequency response. The load resistance is set (R_load = 10 kΩ) when the acceleration ranges are placed under the varying frequency ranges from 320 Hz to 450 Hz. The voltage and displacement acceleration response are an extension of the frequency domain process. This is shown in Figure 14A,B.

Figure 14C portrays the results related to the voltage with acceleration changes for displacement and generated voltages for various geometries simulated as discussed. As per the proposed methodology, the observed voltage varies in terms of acceleration based on the theoretical conditions. Based on the vibration value, the printed model reacted to the fluctuated frequency and reached some acceleration changes. The load area from the designed geometries had the maximum displacement. The graphene material has conductivity; when the structure is subject to vibration, it extends the maximum displacement and makes way for achieving the maximum voltage generation. Based on the simulation in Comsol Multiphysics, it is concluded that the 3D-designed proposed geometric structures are successfully analysed and produce the harvested voltage.

## 5. Conclusions

The study aims to investigate the energy harvesting performance of 3D-printed graphene through experimental investigation and simulation. The research involves various stages, including developing a material using 3D printing techniques and analysing energy harvesting using simulation and experimental methods. The simulation of energy harvesting utilised the unique properties of graphene material and a 3D-designed structure, resulting in a harvested output of 0.69 µV. In the experimental procedure, a vibration-based cantilever beam technique was used, and it was observed that the power and frequency analysis variations in both 3D-printed models were identical. The FDM-based 3D-printed model achieved a power output of 6.4 µW, while the DIW-based 3D-printed model reached a higher harvested power of 12.2 µW. DIW-based structures exhibited better mechanical stability, resulting in exceptional power performance across various resistive load scenarios. In view of the model’s potential application, buildings can utilise this model to energise sensors that manage lighting and optimise heating, ventilation, and air-conditioning (HVAC) systems.

## Figures and Tables

**Figure 1 polymers-16-02397-f001:**
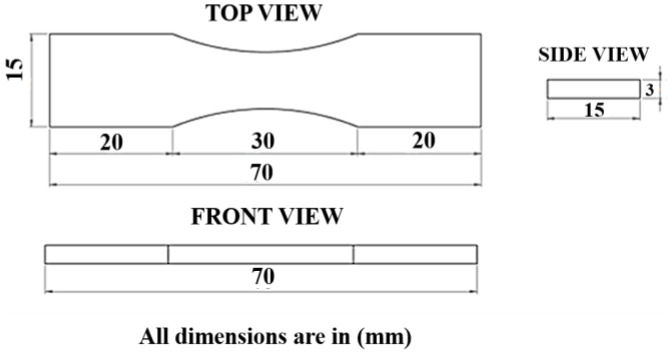
Dimension of the proposed 3D model.

**Figure 2 polymers-16-02397-f002:**
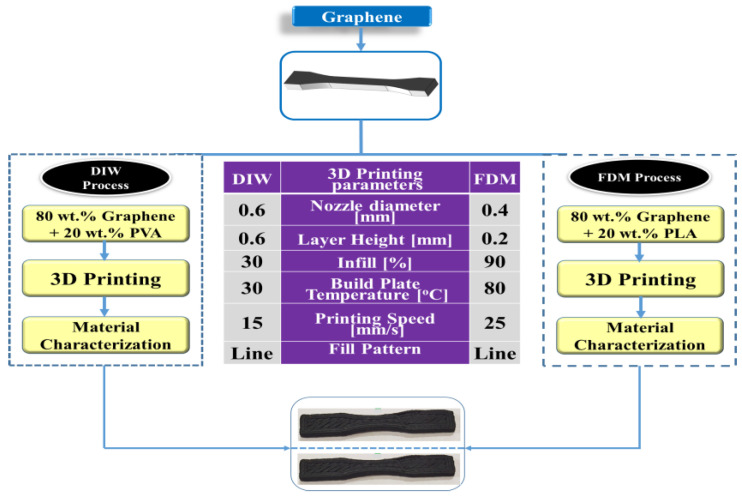
Methodology of extrusion-based 3D printing process.

**Figure 3 polymers-16-02397-f003:**
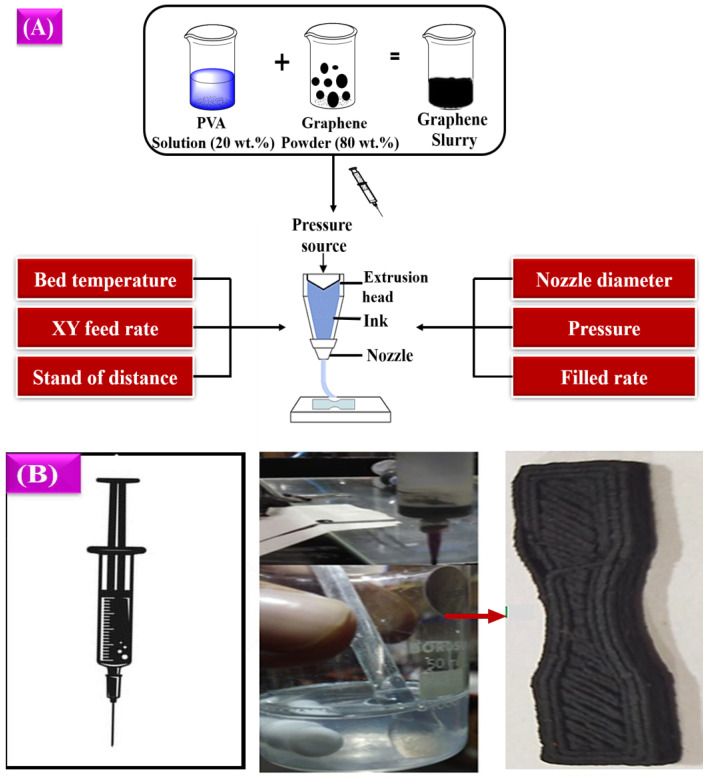
(**A**) Experimental methodology of DIW process and (**B**) printed model.

**Figure 4 polymers-16-02397-f004:**
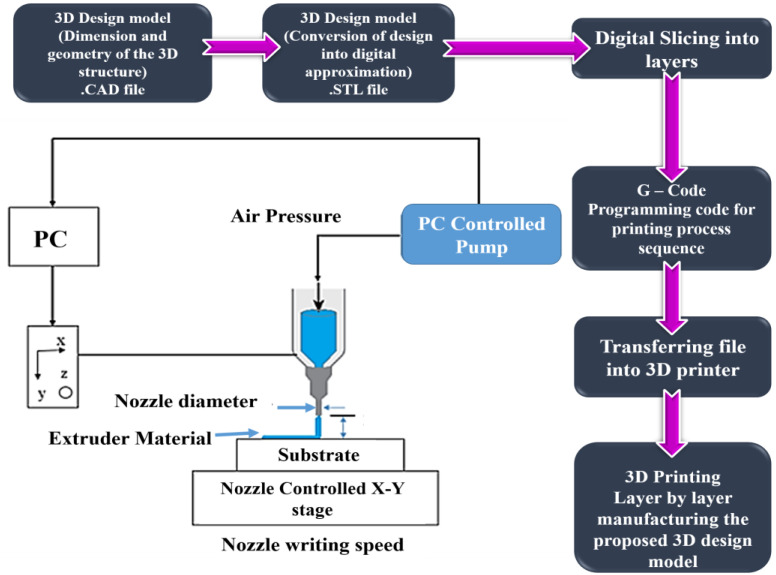
Graphical illustration of FDM-based 3D printing process.

**Figure 5 polymers-16-02397-f005:**
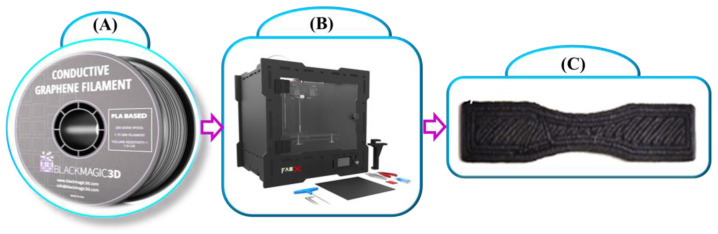
FDM-based 3D printing process: (**A**) graphene filament, (**B**) FDM 3D printer, (**C**) 3D-printed model.

**Figure 6 polymers-16-02397-f006:**
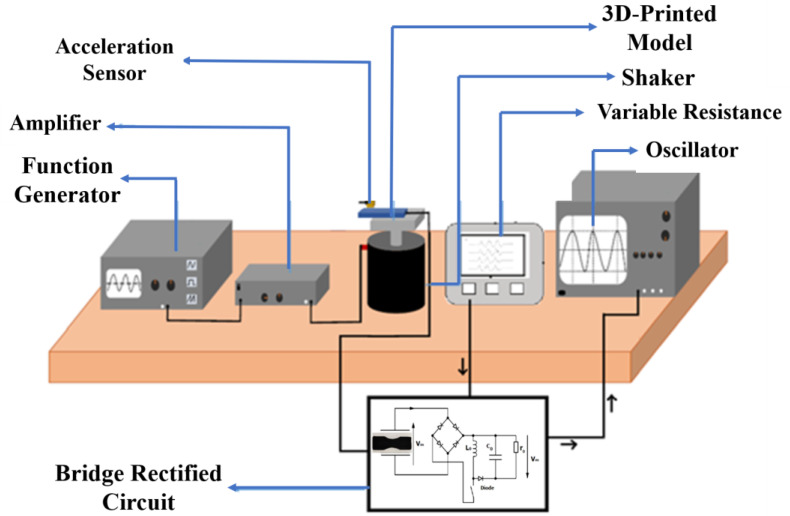
Block diagram for experiment-based cantilever beam technique.

**Figure 7 polymers-16-02397-f007:**
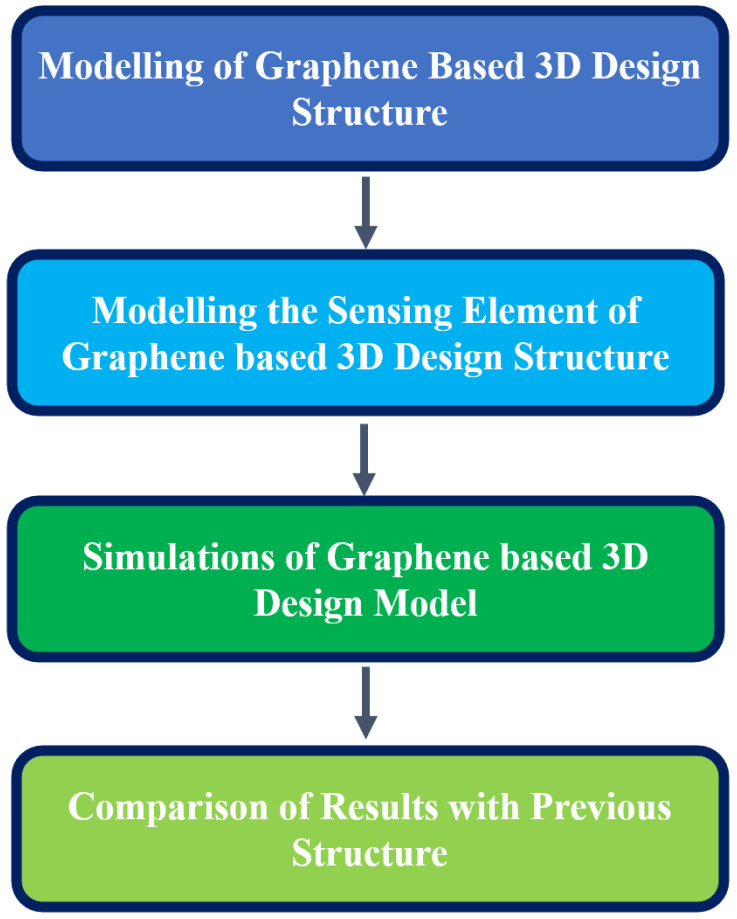
Simulation of graphene-based 3D model.

**Figure 8 polymers-16-02397-f008:**
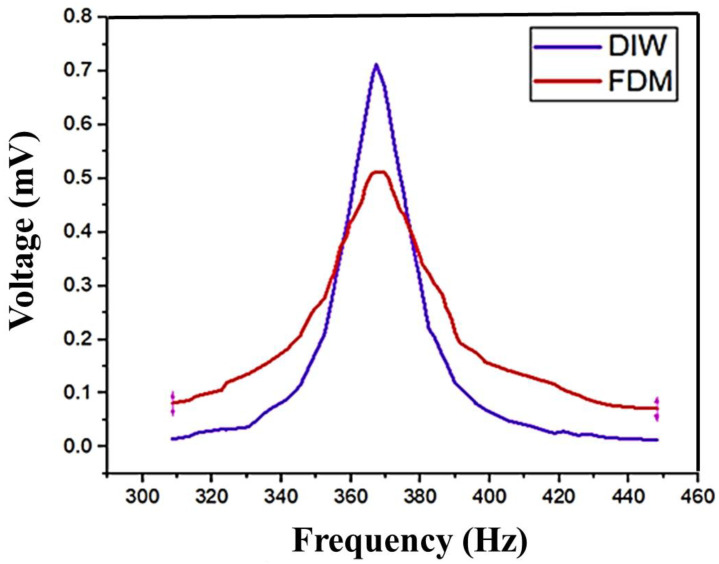
Experimental results for energy harvesting—output voltage results with respect to frequency.

**Figure 9 polymers-16-02397-f009:**
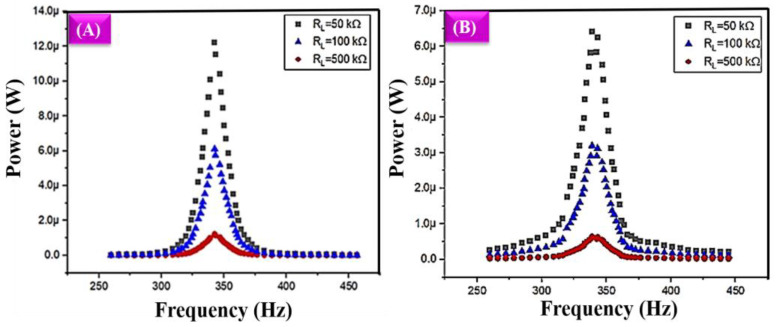
Experimental output of power and frequency changes with various load resistance conditions: (**A**) DIW-based 3D-printed structure, (**B**) FDM-based 3D-printed structure.

**Figure 10 polymers-16-02397-f010:**
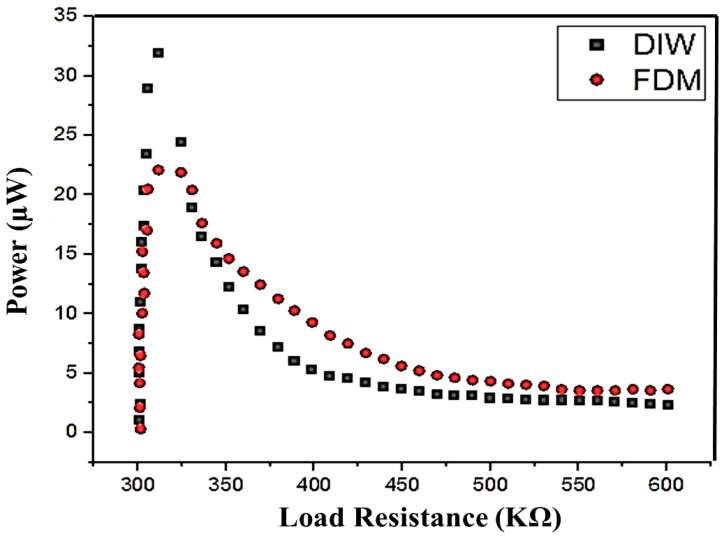
Experimental output of power and load resistance of both DIW- and FDM-based 3D-printed structure.

**Figure 11 polymers-16-02397-f011:**
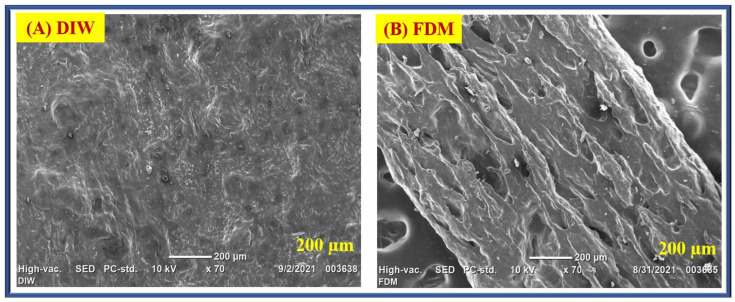
SEM analysis: (**A**) DIW and (**B**) FDM 3D-printed model.

**Figure 12 polymers-16-02397-f012:**
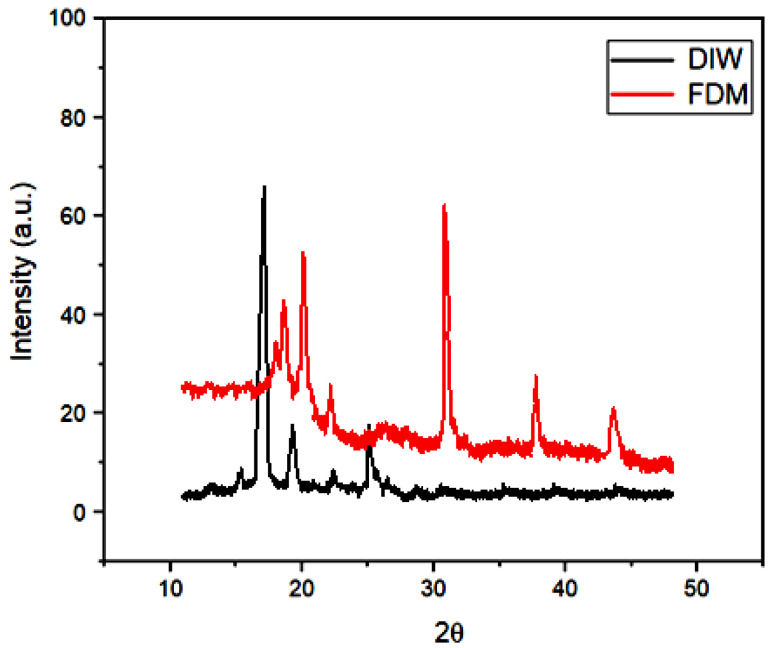
XRD analysis of both FDM and DIW 3D-printed models.

**Figure 13 polymers-16-02397-f013:**
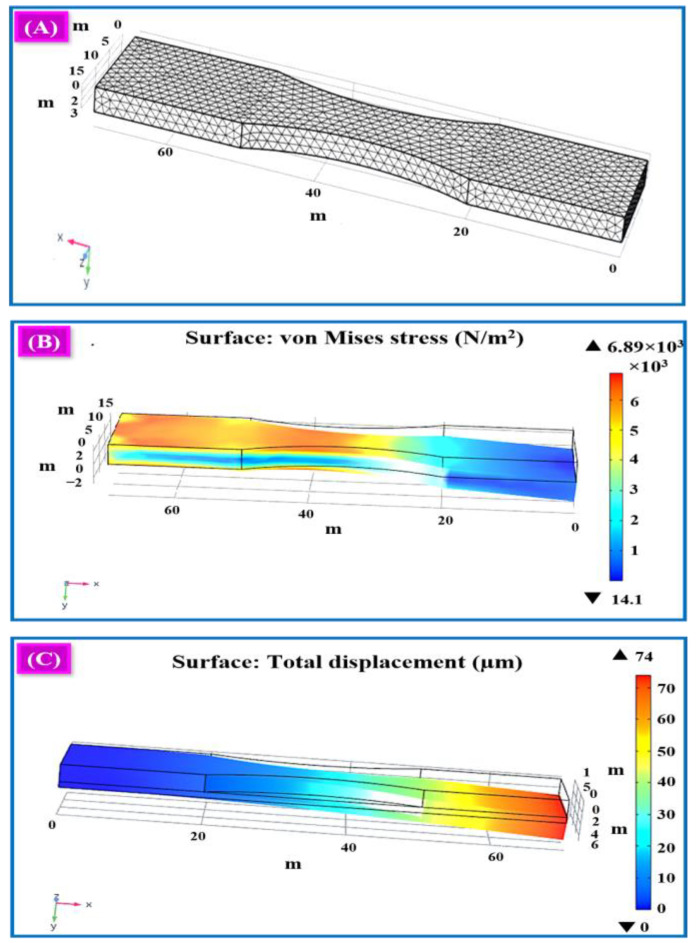
(**A**) Meshing of the 3D-designed sample, (**B**) von Mises stress contour at the resonant frequency of the 3D model, and (**C**) total displacement at the resonant frequency of the 3D model.

**Figure 14 polymers-16-02397-f014:**
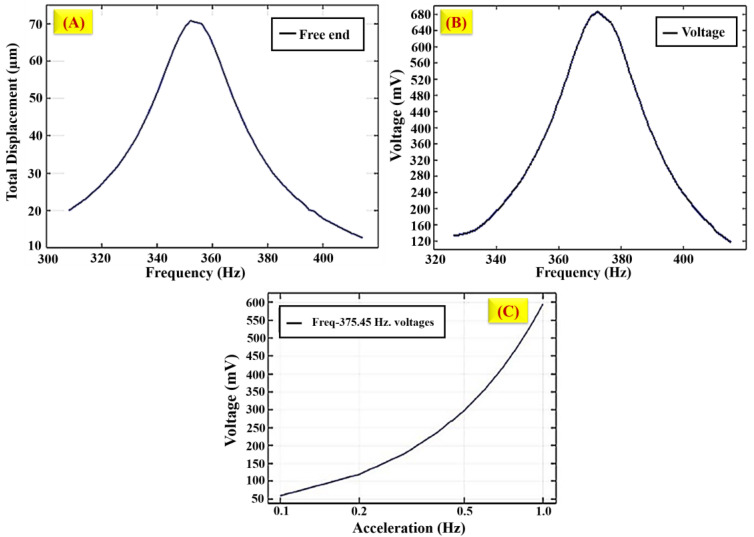
(**A**) Displacement contour at the resonant frequency of 3D model, (**B**) voltage output with respect to frequency range, (**C**) variation in voltage with respect to acceleration.

**Table 1 polymers-16-02397-t001:** Processing parameters for meshing simulation analysis.

Description	Value
Number of vertex elements	16
Number of edge elements	268
Number of boundary elements	3136
Number of elements	16,201
Free meshing time	0.80s
Minimum element quality	0.2611
Maximum element size	0.082
Minimum element size	8.0 × 10^−4^
Curvature factor	0.23
Maximum element growth rate	1.28
Predefined size	Extremely fine

**Table 2 polymers-16-02397-t002:** Analysis of harvested output power.

S. No.	3D Printing Methods	Load Resistance (kΩ)	Frequency (Hz)	Power (µW)
1	DIW	50	309	12.22
2	DIW	100	342	6.11
3	DIW	500	376	1.24
4	FDM	50	303	6.4
5	FDM	100	338.89	3.2
6	FDM	500	372.45	0.6

## Data Availability

The data pertaining to this study have not been deposited in a publicly accessible repository, given that all relevant data are thoroughly detailed in the article or appropriately cited in the manuscript.

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
