# Peer review of "Evaluating the Piezoelectric Energy Harvesting Potential of 3D-Printed Graphene Prepared Using Direct Ink Writing and Fused Deposition Modelling"

_polymers, 2024, doi:10.3390/polym16172397_

Round 1

Reviewer 1 Report (New Reviewer)

Comments and Suggestions for Authors

The manuscript presents research on the potential of 3D printed graphene for energy harvesting, utilizing Direct Ink Writing (DIW) and Fused Deposition Modelling (FDM) methods. The study involves both simulation and experimental analysis, showing DIW-derived prototypes outperforming FDM-derived ones in terms of power output. The study has potential for publication, provided these issues are resolved:

1.       The literature review is comprehensive; however, it would benefit from a more critical analysis of existing methods and their limitations. Specifically, highlight how your approach addresses these gaps.

2.       Use the following papers. Poly (ethylene terephthalate) glycol/carbon black composites for 4D printing. Direct pellet three-dimensional printing of polybutylene adipate-co-terephthalate for a greener future.

3.       Clarify the reasons for choosing specific parameters in the COMSOL simulation. How do these parameters compare with those in similar studies?

4.       Provide more details on the calibration of the cantilever beam setup. How was the setup validated to ensure accurate measurement of the piezoelectric output?

5.       Explain the selection criteria for the graphene materials used. Were different types or grades of graphene considered?

6.       The results are well-presented, but the discussion could be more in-depth. Compare your findings with similar studies quantitatively, not just qualitatively.

7.       Discuss potential reasons for the higher performance of DIW-derived prototypes. Is it due to better material properties, fabrication accuracy, or other factors?

8.       The section on COMSOL simulation results needs to be more integrated with the experimental results. How do the simulation outcomes validate or contrast with the experimental findings?

9.       Suggest practical applications for the harvested energy. How could this technology be scaled up or integrated into existing systems?

Comments on the Quality of English Language

**

Author Response

Response to Comments of Reviewer 1:

The manuscript presents research on the potential of 3D printed graphene for energy harvesting, utilising Direct Ink Writing (DIW) and Fused Deposition Modelling (FDM) methods. The study involves both simulation and experimental analysis, showing DIW-derived prototypes outperforming FDM-derived ones in terms of power output. The study has the potential for publication, provided these issues are resolved.

Comment 1: The literature review is comprehensive; however, it would benefit from a more critical analysis of existing methods and their limitations. Specifically, highlight how your approach addresses these gaps.

Response: The authors agreed with the reviewer’s comments. The primary objective of this study is to investigate how extrusion-based printing methods impact the piezoelectric properties of graphene-based materials. This experimental study focuses on how graphene is printed, taking into account the printing compatibility and manufacturing process of both the direct ink writing (DIW) and fused deposition modelling (FDM) printing processes. Fused Deposition Modeling (FDM) and Direct Ink Writing (DIW) are prominent 3D printing techniques, each with its strengths and limitations. FDM, while accessible and versatile, often suffers from layer lines, inconsistent mechanical properties, and limited material options. Its reliance on thermoplastic filaments restricts its application in certain industries. Conversely, DIW offers greater material flexibility, enabling the printing of complex geometries and functional components. However, it can be slower, more expensive, and requires precise material formulation.

Both FDM and DIW face challenges in achieving high-resolution, isotropic properties and large-scale production efficiency compared to other 3D printing methods like Stereolithography (SLA) and Selective Laser Sintering (SLS). SLA excels in producing highly accurate parts with smooth surfaces, but material choices and post-processing requirements limit it. SLS offers broader material compatibility and build volume but results in lower surface quality. Ultimately, the optimal 3D printing method depends on the specific application, considering factors like material properties, part complexity, production volume, and cost-effectiveness. So, based on this, it is concluded that FDM and DIW have been considered the best ones for providing better results.

Comment 2: Use the following papers. Poly (ethylene terephthalate) glycol/carbon black composites for 4D printing. Direct pellet three-dimensional printing of polybutylene adipate-co-terephthalate for a greener future.

Response:  The authors agreed with the reviewer’s comments. The paper mentioned above is included in the manuscript.

  • Rahmatabadi D, Bayati A, Khajepour M, Mirasadi K, Ghasemi I, Baniassadi M, Abrinia K, Bodaghi M, Baghani M. Poly (ethylene terephthalate) glycol/carbon black composites for 4D printing. Materials Chemistry and Physics. 2024 Jul 17:129737.
  • Karimi A, Rahmatabadi D, Baghani M. Direct pellet three-dimensional printing of polybutylene adipate-co-terephthalate for a greener future. Polymers. 2024 Jan 18;16(2):267.

Comment 3: Clarify the reasons for choosing specific parameters in the COMSOL simulation. How do these parameters compare with those in similar studies?

Response: The authors agreed with the reviewer’s comments. For this analysis, some specific parameters have been chosen for performing meshing analysis, such as a number of vertex elements, edge elements, and boundary elements. Based on the 3D CAD design, this has been analysed. The same set of data has also been verified for vibration analysis. The meshing analysis performed in COMSOL Multiphysics is described below.

Table 1. Processing parameters for meshing simulation analysis.

Description

Value

Number of vertex elements

16

Number of edge elements

268

Number of boundary elements

3136

Number of elements

16201

Free meshing time

0.80s

Minimum element quality

0.2611

Maximum element size

0.082

Minimum element size

8.0E-4

Curvature factor

0.23

Maximum element growth rate

1.28

Predefined size

Extremely fine

Comment 4: Provide more details on the calibration of the cantilever beam setup. How was the setup validated to ensure accurate measurement of the piezoelectric output?

Response: The author agreed with the comments. The cantilever beam setup was calibrated properly before being experimented. The procedure encompasses the establishment of measurement parameters, the calibration of excitation and measurement systems, the optional execution of modal analysis, the assessment of frequency response functions, the creation of calibration curves, and the evaluation of repeatability and accuracy. This methodology guarantees a precise relationship between the input excitation and the resultant measured output response.

Comment 5: Explain the selection criteria for the graphene materials used. Were different types or grades of graphene considered?

Response: The author agreed with the reviewer's comments. High-quality graphene grades were used for this study. This graphene material was purchased from Blackmagic 3D by Graphene 3D Lab, located in New York, United States. Graphene of high quality, which is defined by a low defect density and substantial flake size, generally demonstrates enhanced electrical conductivity, a vital attribute for effective energy conversion. In the context of piezoelectric energy harvesting, graphene exhibiting a high piezoelectric coefficient is preferred. Ultimately, the selection of the most appropriate graphene grade for a given energy harvesting application necessitates a meticulous consideration of these factors alongside the required performance criteria. High-quality graphene is generally favoured for energy harvesting purposes; the specific demands of each application determine the most appropriate grade. Considerations such as the type of energy harvesting (e.g., piezoelectric or thermoelectric), the desired power output, and the feasibility of production scalability are essential when choosing graphene to achieve optimal performance.

Comment 6: The results are well-presented, but the discussion could be more in-depth. Compare your findings with similar studies quantitatively, not just qualitatively.

Response: The authors agreed with the reviewer’s comment. The same set of similar results indicates that the 3D printed graphene has better power generation performance. Likewise, Haque et al. [73] describes the development of a 3D-printed triboelectric device capable of harvesting energy and detecting mechanical deformations with a maximum power density of 10.6 μW/cm2.  Bhavanasi et al. [74] states that bilayer films containing poled PVDF-TrFE and graphene oxide exhibit better piezoelectric energy harvesting performance, exhibiting a voltage and power output of 4.41 W/cm^2. Based on these comparisons, this study has shown better energy harvesting capabilities. Karan describes how the Fe-doped reduced graphene oxide/PVDF nanocomposite film exhibits excellent piezoelectric energy harvesting performance. The primary outcomes measured in this study were the open-circuit output voltage (up to 5.1 V) and short-circuit current (up to 0.254 μA) of the Fe-doped RGO/PVDF nanocomposite film when subjected to repetitive human finger imparting. Kwon states that the Graphene transparent electrodes improve the performance of PZT-based piezoelectric energy harvesters. The primary outcome measured in this study was the performance of the PZT nanogenerator, specifically the output voltage (∼2 V), current density (∼2.2 μA cm−2), and power density of ∼88 mW cm−3.

Comment 7: Discuss potential reasons for the higher performance of DIW-derived prototypes. Is it due to better material properties, fabrication accuracy, or other factors?

Response: The authors agreed with the reviewer’s comments. Direct Ink Writing (DIW) provides enhanced control over the deposition of materials, facilitating the accurate placement of conductive inks. This precision contributes to improved conductivity and reduced variability in the resulting product. Furthermore, DIW is capable of utilizing a broader spectrum of conductive materials, including those that are incompatible with filament extrusion methods used in Fused Deposition Modeling (FDM). Although FDM has made progress in the realm of conductive material printing through the introduction of conductive filaments, it frequently experiences diminished conductivity due to the presence of layer lines and uneven material distribution. In contrast, DIW's capacity to create thicker and more uniform conductive pathways significantly enhances electrical performance.

Comment 8: The section on COMSOL simulation results needs to be more integrated with the experimental results. How do the simulation outcomes validate or contrast with the experimental findings?

Response: The authors agreed with the reviewer’s comments. The main reason to conduct the COMSOL simulation study is to understand “how the energy harvesting output has been derived and whether graphene has produced harvested power or not”. In the COMSOL simulation, idle-grade graphene was used. For Idle graphene, mechanical stability has been considered as a very low value. At the same time, Composite material is also not possible to utilise in the simulation.  So we can’t able to add supporting material like PVA and PLA. So, we can’t able to compare the simulation study with the Experimental study.  

Comment 9: Suggest practical applications for the harvested energy. How could this technology be scaled up or integrated into existing systems?

Response: The authors agreed with the reviewer's comments. Energy harvested from the environment has the potential to supply power to low-energy devices such as Internet of Things (IoT) sensors, wearable technology, and equipment situated in isolated areas. This innovative technology can be seamlessly integrated into a variety of systems. For example, buildings can utilize energy harvesting technologies to energise sensors that manage lighting and optimize heating, ventilation, and air conditioning (HVAC) systems. Similarly, vehicles may incorporate piezoelectric materials within their tires or suspension systems to produce electrical energy. Furthermore, consumer electronics can integrate energy-harvesting components to prolong battery longevity. Although the amount of energy collected is typically modest, enhancements in efficiency, the expansion of harvesting surfaces, advancements in energy storage solutions, and the implementation of intelligent management systems can significantly enhance the practicality and scalability of this technology.

Reviewer 2 Report (New Reviewer)

Comments and Suggestions for Authors

The manuscript entitled “Evaluating the Piezoelectric Energy Harvesting Potential of 3D Printed Graphene Prepared using Direct Ink Writing and Fused Deposition Modelling” investigates the energy harvesting performance of 3D-printed graphene composites using Direct Ink Writing (DIW) and Fused Deposition Modelling (FDM) and compares the results with simulations. There are several areas where the authors could improve the presented work.

1.       The authors write, " However, compared with FDM, DIW voltage output is high." Please provide an explanation for the higher voltage obtained with DIW.

2.       Please use a standard notation for denoting power density throughout the manuscript. Ensure consistency by using either μW/cm² or W/cm². Similarly, modify the representation of resistance from KΩ to kΩ wherever applicable.

3.       Please include a real image of the fabricated device used for energy harvesting measurements.

4.       The authors write, "Due to their PVA content, DIW-based 3D-printed structures have good mechanical stability." However, no mechanical properties of the structures are compared. Please record these properties, provide the data, and discuss them in detail.

5.       In line 434, the authors appear to mistakenly refer to PMMA as the polymer matrix. Please correct this.

6.       In line 122, replace "3D mode" with "3D model."

7.       Label the dimensions of the 3D printed model for both processes utilized (Fig. 2B).

8.       Specify the SEM instrument used for characterizing the surface morphology of the printed samples. Please include this information in section 3.1.

9.       How much load was applied to the devices for evaluating their energy harvesting capabilities?

10.  Discuss the observed power density results of the 3D-printed devices in comparison with those reported in the literature.

11.  What thickness was employed for the DIW and FDM models?

12.  Given that the graphene composite was prepared with PVA for DIW and with PLA for FDM, how do the authors justify comparing these results when the composites are completely different?

Author Response

Response to Comments of Reviewer 2:

The manuscript entitled “Evaluating the Piezoelectric Energy Harvesting Potential of 3D Printed Graphene Prepared using Direct Ink Writing and Fused Deposition Modelling” investigates the energy harvesting performance of 3D-printed graphene composites using Direct Ink Writing (DIW) and Fused Deposition Modelling (FDM) and compares the results with simulations. There are several areas where the authors could improve the presented work.

Comment 1: The authors write, "However, compared with FDM, DIW voltage output is high." Please explain the higher voltage obtained with DIW.

Response: The authors agreed with the reviewer’s comments. Direct Ink Writing (DIW) provides enhanced control over the deposition of materials, facilitating the accurate placement of conductive inks. This precision contributes to improved conductivity and reduced variability in the resulting product. Furthermore, DIW is capable of utilizing a broader spectrum of conductive materials, including those that are incompatible with filament extrusion methods used in Fused Deposition Modeling (FDM). Although FDM has made progress in the realm of conductive material printing through the introduction of conductive filaments, it frequently experiences diminished conductivity due to the presence of layer lines and uneven material distribution. In contrast, DIW's capacity to create thicker and more uniform conductive pathways significantly enhances electrical performance.

Comment 2: Please use a standard notation for denoting power density throughout the manuscript. Ensure consistency by using either μW/cm² or W/cm². Similarly, modify the representation of resistance from KΩ to kΩ wherever applicable.

Response: The authors agreed with the reviewer’s comments. Modified and included in the manuscript.

Comment 3: Please include a real image of the fabricated device used for energy harvesting measurements.

Response: The authors agreed with the reviewer’s comments. This experiment was done in SMART MATERIAL CHARACTERIZATION (SMC) LAB, IIT Madras. We are not supposed to be allowed to take any pictures of the entire setup. However, the printed part has been mentioned in Figures 2B and 4C.

 Comment 4: The authors write, "Due to their PVA content, DIW-based 3D-printed structures have good mechanical stability." However, no mechanical properties of the structures are compared. Please record these properties, provide the data, and discuss them in detail.

Response: The authors agreed with the reviewer’s comments. The statement has been modified and revised from the manuscript. The primary objective of this study is to investigate how extrusion-based printing methods impact the piezoelectric properties of graphene-based materials. This experimental study focuses on how graphene is printed, taking into account the printing compatibility and manufacturing process of both the direct ink writing (DIW) and fused deposition modelling (FDM) printing processes. A binder material is used to provide additional mechanical stability for printing the graphene material. Specifically, the study kept 80 wt.% of the graphene material constant while using 20 wt.% of PVA for the DIW process and 20 wt.% of PLA for the FDM process.

In this study, distinctive binder materials were used for experimental compatibility. For the Direct Ink Writing process, PVA was used as a binder material for graphene-based composites, while PLA was used for the FDM process. In both manufacturing processes, the same weight percentage of Graphene (80 wt.%) was used as the primary material, along with 20 wt.% of PVA for the DIW process and 20 wt.% of PLA for the FDM process. The composition of the 3D-printed sample is detailed in Table 1.

Table 1 Composition of 3D-printed sample prepared using DIW and FDM techniques

S. No

Printing methods

Graphene

Material (wt.%)

Binder material (wt.%)

Remarks

1.

DIW

80

PVA - 20

PVA has been used as the binder material for better printing compatibility.

2.

FDM

80

PLA - 20

PLA has been used as the binder material for better printing compatibility.

Comment 5: In line 434, the authors appear to mistakenly refer to PMMA as the polymer matrix. Please correct this.

Response: The authors agreed with the reviewer’s comments. Modified.

Comment 6: In line 122, replace "3D mode" with "3D model."

Response: The authors agreed with the reviewer’s comments. Modified.

Comment 7: Label the dimensions of the 3D printed model for both processes utilised (Fig. 2B).

Response: The authors agreed with the reviewer’s comments. It's mentioned below and included in the manuscript.

2.1 Proposed Design

The dimensions of the proposed model are shown in Figure 1. A small structure size of about 70 mm in length, 15 mm in width, and 3mm in thickness has been considered for this proposed work.

Figure 1. Dimension of the proposed 3D model.

Comment 8: Specify the SEM instrument used for characterising the surface morphology of the printed samples. Please include this information in section 3.1.

Response: The authors agreed with the reviewer’s comments.  For this experiment, we have used the FEI-Quanta FEG 200F SEM machine (USA) at IIT Madras with the following specifications,

Specifications:

Source: FEG assembly with Schottky emitter (-200 V to 30 kV)

Beam Current: >100 nA

Resolution: 1.2 nm (Gold nanoparticles suspended on Carbon substrate)

Magnification: 12X-105X

Mode: High Vacuum (for conductors), Low Vacuum (for insulators) and ESEM (for biospecimen)

Accessories are available: EDX and WDS.

Comment 9: How much load was applied to the devices to evaluate their energy harvesting capabilities?

Response: The authors agreed with the reviewer’s comments.  This study of power and frequency changes with various load resistance conditions. In this analysis, 50 kΩ, 100 kΩ, and 500 kΩ have been used for the study. It shows the power and frequency analysis with various resistive load conditions of the DIW and FDM-based 3D printed model.

Comment 10: Discuss the observed power density results of the 3D-printed devices in comparison with those reported in the literature.

Response: The authors agreed with the reviewer’s comments.  The FDM-based 3D printed model achieved a power output of 6.4 µW/cm2, while the DIW-based 3D printed model reached a higher harvested power of 12.2 µW/cm2. Haque et al. [73] describes the development of a 3D-printed triboelectric device capable of harvesting energy and detecting mechanical deformations with a maximum power density of 10.6 μW/cm2.  Bhavanasi et al. [74] states that bilayer films containing poled PVDF-TrFE and graphene oxide exhibit better piezoelectric energy harvesting performance, exhibiting a voltage and power output of 4.41 μW/cm2. Based on this comparison, this study has shown better energy harvesting capabilities.

Comment 11: What thickness was employed for the DIW and FDM models?

Response: The authors agreed with the reviewer’s comments. As per the literature, the thickness was used to be 3mm for both DIW and FDM models. The diagram is also mentioned in comment section 7.

Comment 12: Given that the graphene composite was prepared with PVA for DIW and with PLA for FDM, how do the authors justify comparing these results when the composites are completely different?

Response: The authors agreed with the reviewer’s comments. The primary objective of this study is to investigate how extrusion-based printing methods impact the piezoelectric properties of graphene-based materials. This experimental study focuses on how graphene is printed, taking into account the printing compatibility and manufacturing process of both the direct ink writing (DIW) and fused deposition modelling (FDM) printing processes. A binder material is used to provide additional mechanical stability for printing the graphene material. Specifically, the study kept 80 wt.% of the graphene material constant while using 20 wt.% of PVA for the DIW process and 20 wt.% of PLA for the FDM process.

In this study, distinctive binder materials were used for experimental compatibility. For the Direct Ink Writing process, PVA was used as a binder material for graphene-based composites, while PLA was used for the FDM process. In both manufacturing processes, the same weight percentage of Graphene (80 wt.%) was used as the primary material, along with 20 wt.% of PVA for the DIW process and 20 wt.% of PLA for the FDM process. The composition of the 3D-printed sample is detailed in Table 1.

Table 1 Composition of 3D-printed sample prepared using DIW and FDM techniques

S. No

Printing methods

Graphene

Material (wt. %)

Binder material

(wt. %)

Remarks

1.

DIW

80

PVA - 20

PVA has been used as the binder material for better printing compatibility.

2.

FDM

80

PLA - 20

PLA has been used as the binder material for better printing compatibility.

Round 2

Reviewer 1 Report (New Reviewer)

Comments and Suggestions for Authors

Accept.

Reviewer 2 Report (New Reviewer)

Comments and Suggestions for Authors

The authors have addressed the reviewers' comments, and the manuscript is now suitable for acceptance in its current form.

This manuscript is a resubmission of an earlier submission. The following is a list of the peer review reports and author responses from that submission.

Round 1

Reviewer 1 Report

Comments and Suggestions for Authors

In this paper, the authors explore the 3D printing method using Direct Ink Writing (DIW) and Fused Deposition Modelling (FDM) to fabricate PVA-graphene composites for energy harvesting. Both simulation and experimental techniques were used to analyze energy harvesting. The paper is not recommended for publication as there are major concerns in the approach that has been followed, as detailed below:

Major Concerns

1)     The basic goal of this work was to compare samples made from two different material systems. When comparing two polymer composite systems, a few elements must be common, like the polymer utilized, the quantity of filler material employed, the printing conditions, and so on. In this study, they were all distinctive.

·       The polymer used in the DIW is PVA, while PLA was utilized in the FDM.

·       The filler contents in the investigation were also not kept consistent since the filler contents in the commercial filament are unknown. To provide a fair comparison, it is preferable to disperse the filament with certain chemicals and estimate the filler content, which may then be employed in the DIW.

Therefore, maintaining consistent printing conditions is crucial since they considerably impact material characteristics. Discrepancies between these factors will not make the comparison fair.

2)     The author claims, “Due to their PLA composition, the DIW-based structures exhibited commendable mechanical stability, resulting in exceptional power performance across various resistive load scenarios.” However, it was clear from the methodology that the authors used PVA as the polymer in the DIW, thus, the conclusion from this statement is weak and not objective.

3)     The paper overall is weakly written and the English needs to be enhanced significantly.

Minor Comments

·       The authors stated, “The DIW SEM image shows that the PLA has a smaller particle size and formed aggregates that are equally spread in the entire surface area.  indicates that fewer stacks exist and that the structure has been exfoliated on graphene. The FDM SEM image shows that the PVA with graphene has a rippled surface.” The approach shows that the authors utilized PVA as a polymer for DIW and PLA as a polymer for FDM. Hence, the above-mentioned statement is not accurate and should be corrected.

·       The first paragraphs up to Page 3 in the Introduction section is long and provide well-known and general information. It is recommended that these paragraph are shortened significantly. Also, no need for Tables 1 and 2 as this is well-known and the authors can just refer to review papers on these topics. In fact, the Introduction section should be re-written completely as only the last paragraph contains information directly related to the paper.

·       The classification and the grouping of 3D printing techniques mentioned in the Introduction section have to be corrected. Kindly refer to ASTM 52900 for classifications and sub-classifications of 3D printing techniques.

·       Two compositions were considered for fabricating PVA-graphene samples, and only one was selected for comparison. There is no strong justification for choosing the composition of PVA-graphene preparation.

·       The presentation of the figures should be improved significantly.

·       The placement of citations, especially in Table 2, has to be corrected.

Comments on the Quality of English Language

The paper overall is weakly written and the English needs to be enhanced significantly.

Reviewer 2 Report

Comments and Suggestions for Authors

The manuscript submitted by this author reports a method for printing graphene materials using 3D printing, specifically direct ink writing and melt deposition modeling, resulting in superior power performance. However, the following major revisions should be resolved prior to publication.

 1.      In the manuscript, the resolution of figures (Page 2, Figure 6; Page 8, Figure 3) are poor. This experimental method diagram needs to be improved, and the images should have standardized labels and annotations

2. In the manuscript "Experimental procedure for extrusion-based 3D printing method" (especially in Part 2.1, Direct ink writing method), the description of the preparation method process is very redundant and makes it difficult to understand the key point. The author needs to carefully simplify and modify the text.

(1): In this part: a) the preparation of PVA solution, here is just a simple physical mixing, why spend so much space with graphics to describe it is not necessary? The authors mentioned that "the output quality really does not meet the expected requirements", which leads to poor results with a lack of specific images and data. The authors need to provide additional explanations for this.

(2): In this part: b) preparation of graphene paste, I am curious about how the author explores the solution of graphene powder without PVA ratio, as the relevant information is not provided in the manuscript.

3. Author needs to normalize and modify the manuscript structure and add "results" and "discussion" sections of the manuscript, such as SEM and piezoelectric performance tests, to the second part of the manuscript.

4. The manuscript lacks an adequate description of the material itself, and only the characterization of morphology and structure is not enough, the author should provide more details and relevant characterization discussions, including functional groups and other material components.

5. The results of the analysis of the energy collection characterization of materials, such as the relationship between frequency and output voltage, frequency and power, are not enough, and only stay at the data level, without interpretation from a theoretical perspective. This author should provide an external explanation of the piezoelectric effect, such as the relationship between frequency and output voltage: an explanation of the reason for this tendency to change is the deformation that occurs from the perspective of the piezoelectric material and conductive XRD testing to complement the material's crystal structure information.

6. In the manuscript (page 17, line 414), the authors mention that 3D printed graphene has better power generation properties. However, the authors only compared the results of their own experiments, not those of peers. The authors need to supplement the comparative results, otherwise, the statement is not rigorous and reliable.

Comments on the Quality of English Language

Authors need to carefully compress and revise the text.

Reviewer 3 Report

Comments and Suggestions for Authors

This article attempts to 3D print parts with graphene infills for energy harvesting purposes. The idea of 3D printing polymer with graphene for energy harvesting is not new. State of the art work including 3D print PVDF with Graphene infill for significantly better piezoelectric properties. The logic for this article needs to be improved. The suggestion is reject.

1.      In your introduction, the piezoelectric property of graphene and the physics behind it is not well described. Please add reference and explanation. Below is one suggestion.

a.      “Xu, K., Wang, K., Zhao, W. et al. The positive piezoconductive effect in graphene. Nat Commun 6, 8119 (2015). https://doi.org/10.1038/ncomms9119”

2.      The purpose of using PLA and PVA is not clear. Please add explanation.

3.      Section 2.1.1 first paragraph, please explain what is a “correct proportion”, and how you review and determine your iterations.

4.      Explain since you have found the “correct proportion”, why do you still have two proportions (2 g PVA+ 18g H2O, and 3g PVA + 17g H2O)?

5.      “Finally, 4.5 grams of the PVA solution (60%) was mixed with 1.5 grams of graphene powder (40%) to prepare the slurry.” and “A combination of PVA solution (20%) and graphene powder (80%) was prepared for final printing.” conflicts with each other. Please explain or correct.

6.      Most of your figures are quite blurry, please replace with higher resolution ones.

7.      “The optimised output model is achieved.” How do you optimize your model? Explain it. Also, correct the typo “optimised”, and run a grammar check.

8.      “The slurry of the two solutions was good, and the printing also had proper dispersion”. If you mean that the distribution of the PVA and graphene in the mixture is uniform and stable based on your observation, describe your observation, and mention if precipitation can be observed after a certain time.

9.      In 2.1.2, after you print the sample, how do you dry it? Does PVA act as a glue for the graphene and the water just evaporate? How long does it take to dry? Did you used any heating source?  

10. In section 2.2, please include the volume percent of graphene in your filament.

11. In section 2.2, “Size: 100 grams”, correct it. 

Comments on the Quality of English Language

Must be improved.